# Measurement and Function of the Control Dimension in Parenting Styles: A Systematic Review

**DOI:** 10.3390/ijerph16173157

**Published:** 2019-08-29

**Authors:** Marta González-Cámara, Alfonso Osorio, Charo Reparaz

**Affiliations:** 1School of Education and Psychology, University of Navarra, 31009 Pamplona, Spain; 2Institute for Culture and Society, University of Navarra, 31009 Pamplona, Spain

**Keywords:** parenting styles, behavioral control, coercion, strictness, supervision, systematic review

## Abstract

Recent studies have shown different results in identifying which parenting style is the most beneficial for children, which has encouraged certain authors to wonder whether parental control is still needed for optimal parenting. As such investigations have been conducted with different measuring instruments, it is necessary to check whether the use of different instruments leads to different results. In order to figure this out, a systematic review of the recent literature (Web of Science and Scopus, 2000–2017) was carried out. This review found that, using certain instruments, parental control is associated with better outcomes in children, while using certain others, control is associated with worse outcomes. The difference seems to be in the way of measuring parental control.

## 1. Introduction

Adolescence is a period that involves many personal changes that give rise to numerous family conflicts. When this time comes, parents and children try to strike a balance between their children’s need for independence and their need to maintain control over the welfare of their adolescent children [1]. For this reason, the influence of parenting styles on adolescent development has been frequently studied.

Within the framework of parenting styles, there is an extensive bibliography. Most authors, following the pioneers in this field [2,3], use ‘love/affection’ and ‘control’ as the main dimensions. They consider these dimensions as orthogonal, and their combination leads to 4 different parenting styles: authoritative (high affect and control), permissive (high affect and low control), authoritarian (low affection and high control) and neglectful (low affection and control). Generally, there is some coincidence in the definition of ‘affection’, and authors consensually believe that it is crucial that parents show care, support, understanding, and affection to their children [4,5,6,7]. This dimension has been defined and measured in similar ways, and most studies show that parental warmth is beneficial for children. However, this agreement does not seem to be so clear regarding parental control [8].

Early in the study of parenting styles, some authors [9,10,11,12,13,14] started to make a distinction between two types of parental control: ‘behavioral control’ and ‘psychological control’ The first type of control is associated with parental monitoring, demandingness and supervision practices, that is, parents try to modulate their children’s behavior by establishing certain rules and limits [15,16,17,18,19,20,21,22,23]. This is how the demandingness dimension was described by the most cited authors [2,3,16].

The second type of control (psychological control) refers to parents that try to control their children’s behavior through intrusive parenting practices such as overprotection or control through guilt [11,24,25]. 

There are other authors who do not differentiate between these two types of control. In some of these cases, authors understand parental control as overprotection, coercion, or physical punishment. This kind of control might be somewhere between psychological and behavioral control, but this issue is rarely discussed [26,27,28,29,30].

It is important to note that the control dimension proposed by the original authors in the field was clearly behavioral control, and had no relation with coercion or intrusiveness:‘The term parental control refers to (…) those parental acts that are intended to shape the child’s goal-oriented activity, modify his expression of dependent, aggressive, and playful behavior, and promote internalization of parental standards. Parental control as defined here is not a measure of restrictiveness, punitive attitudes, or intrusiveness’ [2].‘Authoritative Parenting (…): Firm enforcement of rules and standards, using commands and sanctions when necessary (…). Encouragement of the child's independence and individuality’ [3].

Currently, this difference in the definition of ‘parental control’ affects the method of measuring the ‘control’ dimension in the parenting styles questionnaires and thus in the research results, so it is essential to take this issue into account.

The need to consider these differences is especially relevant when trying to compare the impact of parenting styles in different countries. Otherwise, different results between two studies might be wrongly interpreted as a difference between two countries or cultures. For example, Calafat and collaborators [31], after reviewing the variety of results found in different countries and cultures, expected to find the authoritative parenting style more beneficial in European countries, with the exception of the Southern countries, where they hypothesized that the permissive style would get the best outcomes. In their results, however, they found that, in both Southern and Northern European countries, the permissive style yielded similar or better outcomes than the authoritative style. The explanation of this phenomenon might not be in the culture, but perhaps in the way of measuring the ‘control’ dimension in the instrument used. In order to know whether the type of instrument is having an impact on the study’s results, it would be good to carry out that same research but using another questionnaire that measures control as supervision and monitoring and not as restriction and coercion.

A recent meta-analysis does an important job in comparing, across different cultures, some outcomes (behavior problems and academic achievement) of the different parenting styles [32]. The article claims that the studies included had assessed parenting styles defined by Maccoby and Martin [3]. However, this was not exactly the case. Some of the included studies used instruments where the control dimension measures coercion or psychological control, which do not permit a classification according to Maccoby and Martin [3]. Some other systematic reviews and meta-analyses have explored the impact of different parenting variables on delinquency [33], academic achievement [34], anxiety [35], cyberbullying [36], or internalizing behaviors [37]. However, several issues do not permit to clarify our topic. First, these studies do not focus on the instruments used. Furthermore, most of them do not distinguish between different forms of parental control, and some do not permit to see possible differences across countries.

It is therefore important to clarify the terms so as to not cause confusion in the scientific community and in society in general. This is why we have carried out a systematic review of the literature in order to (1) know the different ways of measurement that have been used between 2000 and 2017 to evaluate parenting styles; (2) know which emotional and behavioral aspects of adolescents’ development have been taken into account more frequently as outcomes and (3) estimate the association that exists between the ways of measurement found and the adolescents’ emotional and behavioral development.

We expect to find different results depending on the type of parental control each study measures. 

## 2. Materials and Methods 

### 2.1. Search Criteria

A systematic search was carried out in mid-2018 on the Web of Science and Scopus databases using the syntax shown in Table 1. As the conceptual problem above mentioned about parental control is a contemporary issue and it seems that the discussion has been increasing since the beginning of the 21st century, the search was limited to publication years 2000 to 2017. Additionally, the search was limited to articles written in English or Spanish for the following reasons: (1) they are the most commonly used languages among the studies found; (2) the main expected differences regarding the correlates of the control dimension are between English- and Spanish-speaking countries; and (3) these are the languages we authors are confident in.

### 2.2. Inclusion and Exclusion Criteria 

Regarding methodology, we classified the inclusion criteria in three aspects:

(a) Study design. For this review, the studies had to include original empirical research (reviews and meta-analyses were excluded); parenting styles (including measures of affect and control) had to be assessed through questionnaires (interviews and observational assessment were excluded, as well as studies measuring affect but not measuring control); and parenting had to be linked to outcomes (descriptive studies of prevalence of parenting styles were excluded).

(b) Informants. We considered only those studies in which children answered about their parents’ practices towards them in the present, that is, informant children (studies with reports from parents or other people, or with reports from adult children remembering their adolescence, were excluded). Since instruments for different informants are usually different, we focused on the most used type.

(c) Population. As we have a special interest in knowing the association between parenting styles and adolescents’ development, only those articles focusing on adolescents were included. Studies had to include participants aged between 13 and 18 years. Furthermore, when other ages were additionally included in one study, ages over 21 or below 9 could not be included: if such ages were included, the study was excluded from the review. 

Concerning the topic, our goal was to discover the association between the way of measuring parenting styles and adolescent development. Since our interest was in the direction of the association (benefit or detriment for adolescent development), we also excluded studies where the outcomes couldn’t be clearly classified as positive or negative (diet, sports, political beliefs...). We needed outcomes with a generalized consensus regarding their positive or negative value. For example, substance use, anxiety, or depression are clearly valued as negative outcomes, while life satisfaction and academic achievement are considered positive. This is not the case regarding some other outcomes, such as diet, sports, or political views (there is no clear agreement regarding the benefit of being vegetarian, playing football, or being libertarian). We cannot classify these outcomes as positive or negative and therefore they are not useful for our study. For this reason, articles using this type of outcomes were excluded from our review.

### 2.3. Selection Process

After completing the search process, all documents were reviewed, and duplicate articles were identified and eliminated using the reference management software Mendeley (Version 1.19.4, Elsevier, Amsterdam, The Netherlands). Next, following the inclusion and exclusion criteria mentioned in the previous section, we carried out the selection of articles. A first selection was performed reading titles and abstracts, and a second one with the full text.

### 2.4. Analysis

In each study, we first identified which instrument(s) had been used to assess parenting styles. Then we selected the instruments used in at least 5 studies, and analyzed such instruments, paying special attention to the control dimensions.

In the studies that used those instruments, we identified the outcomes evaluated, and their association with the control dimension. We focused on whether there was a significant association and on whether it was beneficial or detrimental to the adolescent, but not on effect sizes. When these results were different for different sub-samples (e.g., by adolescent’s sex or age, or by parent’s sex), the fact was also reported.

## 3. Results

The search in the databases produced the following results (2000–2017): Web of Science: 689 results; Scopus: 734 results. After detecting and eliminating the duplicates, there were 895 publications. After reading the title and summary, 382 documents were excluded. The remaining 513 documents were reviewed in full text. Finally, 163 results were included in the systematic review. Figure 1 shows the flow diagram, based on the PRISMA statement [38].

Out of the 163 articles included in the systematic review, we extracted the information regarding the type of instrument, dimensions and outcomes considered in the research. The results were as follows:

### 3.1. Instruments Used

We registered 32 different scales and questionnaires to measure parenting styles and we focused on the instruments used most frequently; in particular, those used in at least 5 articles. We identified, thus, 7 instruments. However, one of them (Parenting Style Scale) [39] is in Turkish, and we have not been able to access a version in English or Spanish. Another instrument (Parental Authority Questionnaire) [40,41] does not have any specific scale of parental control or authority. We, therefore, kept 5 instruments that met the criteria sought, used in 5–26 articles each (52 articles in total) (Table 2). Next, we will describe each one of these instruments.

### 3.2. Parenting Styles Dimensions

All the instruments used measure the way in which parents give support and affection to their children considering it a necessarily positive dimension. They call it affection/support/involvement/acceptance (ex.: ‘my mother/father makes me feel wanted and needed’ or ‘I can count on my parents to help me out if I have some kind of problem’). However, that similarity that exists in the measure of affect is not observed in the measure of control, thus observing different typologies:-PSI (Parenting Styles Index) and EEEP (Escala para la Evaluación del Estilo Parental) differentiate between behavioral control and psychological control. The items designed to measure behavioral control evaluate the extent to which parents set limits and rules for their children, how they enforce these norms, and to what extent they are aware of their children’s activities (monitoring) (ex.: ‘My mother/father really expects me to follow family norms’ or ‘when I do something wrong my mother/father does not punish me’). The psychological control items have the aim of knowing what level of psychological autonomy parents allow their children, how intrusive they are in their development, and to what degree they use guilt to control their children behavior (ex.: ‘my parents act cold and unfriendly if I do something they don’t like’ or ‘my mother/father makes me feel guilty when I do not do what he/she expects’). The PSI scale considers psychological control and psychological autonomy as two opposite poles of the same construct, however the EEEP distinguishes the scale of promotion of autonomy from psychological control in order to know to what extent parents promote the autonomy of their children and encourage them to have their own ideas and make their own decisions (ex.: ‘she/he encourages me to say what I think even if he/she disagrees’). In most studies with these instruments, when parenting styles are built as categories, the scales used are involvement and behavioral control (not psychological control). Two studies [42,43] use cluster analysis, where categories are more difficult to describe.-ESPA-29 (Parental Socialization Scale) and PARQ/C (Parental Acceptance-Rejection/Control Questionnaire) do not include two different dimensions as the two above mentioned scales do, that is, they do not have one dimension for behavioral control and another one for psychological control, but one single control dimension. In ESPA-29, this dimension is called ‘strictness/imposition’, and it includes items such as: ‘my parents scold me’, ‘my parents hit me’ or ‘my parents take something away from me’ in cases of disobedience. It is a dimension that differs from what others call behavioral control, and is sometimes called Coercion/Imposition [44,45]. The PARQ/C measures permissiveness/strictness (also called Control), with items such as: ‘my mother/father is always telling me how I should behave’ or ‘my mother/father lets me do anything I want to do’. This dimension is more similar to behavioral control, but almost all items have a hint of exaggeration: ‘my mother/father tells me *exactly what time* to be home when I go out’; ‘my mother/father believes in having *a lot of rules* and sticking to them’. In both instruments, there is evidence of orthogonality between the control and the love/affection dimensions [46,47]. The studies that use these instruments, when constructing the parenting styles, use this strictness/imposition/control dimension, together with that of affection.-The structure of the CRPBI (Child’s Report of Parental Behavior Inventory) questionnaire is more complex. Although one of the main axes is autonomy/control, there is no scale (or set of scales) that measures that axis. Instead, the instrument is divided into molar dimensions. On the autonomy side, there are 3 molar dimensions, depending on how autonomy is combined with the poles of the other axis (love-hostility). These dimensions are: autonomy, hostility and autonomy, and autonomy and love. On the control side, we have: control, control and hostility, and love and control. In turn, these molar dimensions are divided into concepts (or sub-scales). The studies with this instrument use different versions: each study focuses on some of the subscales of the original CRPBI, and it is difficult to indicate in each case what kind of control is being measured.

### 3.3. Most Frequently Used Outcomes

Following a suggestion found in Darling [16], the outcomes of each investigation have been classified into 2 groups (Table 3). A first category of outcomes refers to behavioral aspects of the children’s development; a second group includes outcomes related to emotional facets.

The most frequently assessed behavioral outcomes in our review are those related to substance use, prosocial-antisocial behavior and school performance. Each of these variables have been measured in different ways.

The emotional outcomes most often taken into account are those associated with self-concept, self-esteem, and self-efficacy, as well as aspects related to the level of psychological adjustment (depression, anxiety, emotional stability). Some studies have also paid attention to ‘life satisfaction’.

### 3.4. The Role of the Control Dimension in Outcomes

We analyzed the information provided by the studies in which any of the measuring instruments classified as ‘most frequent’ were used. There are thus 52 articles that use any of the 5 scales analyzed. The detailed classification of the articles can be found in Table 4.

In some of these investigations, analysis is made according to types or styles (they show results about which parenting style is the most beneficial). In these cases, we have focused on the comparison between the authoritative and the indulgent/permissive parenting styles. These styles resemble in that parents show high levels of affection but differ in the degree of control. Often one of these two styles is the one that appears as the most beneficial, and for that reason we wanted to check if this depends on the instrument used.

In other studies, the authors run analysis by dimensions (determining if a dimension is associated with better or worse outcomes). In these cases, we have looked at the control dimension, and its association with outcomes. In the case of instruments that measure separately behavioral control and psychological control, we have focused on behavioral control.

Finally, some articles run different analyses (for example, cluster analysis), not offering a clear association between the parental control dimension and the outcomes.

Table 5 summarizes the mentioned results. Out of a total of 26 articles that used the PSI, 21 obtained homogeneous results in favor of the authoritative parenting style, and 5 in favor of the behavioral control dimension (3 of these articles were included in both counts because they did both types of analysis). This is true both in articles that measured behavioral and emotional outcomes. The remaining 3 articles did not obtain significant results in the comparisons we are studying.

Regarding the 5 investigations carried out with the EEEP, 2 of them did cluster analysis, and the differences between the groups could not be attributed to a specific dimension. The other 3 articles performed dimensional analysis, and 2 of them found that control is significantly associated with positive outcomes.

Regarding the CRPBI scale, two articles analyzed the outcomes by styles, and one of them concluded that the authoritative parenting style was associated with greater peer attachment, greater academic self-efficacy, and lower aggressiveness. Three other articles made dimensional analysis, and all of them concluded that parental control was significantly associated with worse outcomes (such as alcohol consumption or depressive symptoms). Another research was not included in our results analysis because it did not explain what dimensions were used to create parenting styles.

The PARQ/C scale was used in 5 of the studies analyzed. Out of the 4 that did analysis by styles, 3 articles showed results in favor of the permissive parenting style in terms of self-esteem or academic performance. The remaining article made a dimensional analysis and found that control was associated with less alcohol consumption.

Finally, the 10 studies that used the ESPA-29 scale had results that are opposite to those using the PSI. Specifically, 8 articles analyzed their results by styles, and 4 of them found that the permissive was more beneficial than the authoritative (the other 4 did not find significant differences between both groups). Furthermore, 2 articles run dimensional analyses, both of them finding that control was associated with negative outcomes. Most of the outcomes with significant associations were emotional (mainly self-esteem), while most behavioral outcomes (bullying, sexist attitude, average grades and disruptive behavior) obtained no significant results (the exception was the outcome ‘aggressiveness/hostility’).

## 4. Discussion

According to our findings, the association between parental control and outcomes depends largely on the instrument used to asses control. In one extreme, we have the PSI and the EEEP, where parental control seems to be clearly a protective factor. With the PSI, the evidence in favor of control is overwhelming. In the case of EEEP, the number of studies with relevant results is lower, but the tendency is the same. In both cases, no study shows an association of parental control with worse outcomes. Parental behavioral control, at least as measured with these instruments, seems to be a positive influence on adolescents.

On the opposite side, we find the ESPA-29. All the studies with significant results find that parental control is associated with poorer outcomes in adolescent children. Since this instrument assesses coercion, it seems that this type of control is detrimental to children’s development.

Somewhere in the middle, we find the CRPBI and the PARQ/C. In both cases, we found 3 articles showing a negative association between parental control and outcomes, and 1 article showing a positive association. The CRPBI has many different sub-scales, and each article uses them in different ways. In some cases, it is not possible to know which items the authors have used, and whether the control they measure can be called strictness, supervision, coercion or otherwise. The PARQ/C, on the contrary, is homogeneously used but, as we have seen, its Control subscale seems to be an extreme behavioral control, and that would explain the majoritarian negative outcomes in the research with this instrument.

Some authors have suggested that the diversity of results might be due to cultural differences. Specifically, it has been claimed that parental control might be negative in Spain, in Southern Europe and in Latin America [27,29,30,31]. However, our findings do not support this hypothesis (see Table 4). The positive correlates of behavioral control found with the PSI were obtained not only in the US and Canada, but also in Latin America (Brazil), in Southern Europe (Spain and Italy), in other European countries (Germany, Iceland, and the Netherlands) and in other world regions (China, India, Iran and Nigeria). On the contrary, the ESPA-29 showed negative results for coercion, and these were found in Spain, Brazil, and the US. The EEEP was used in Spain only, and therefore cross-cultural inferences cannot be done, but the studies show that behavioral control has positive correlates in this country (which the mentioned studies deny). The CRPBI and the PARQ/C were used in studies with different populations, and with opposed results; however, there is no relation between the populations studied and the direction of results. Therefore, it does not seem that the results depend on the country, but on the way the instrument measures control.

Also, it has been said that the control dimension has a different impact on different types of outcomes. Specifically, control is supposed to have a broader impact on behavioral outcomes than on emotional outcomes [16]. Our findings partially support this statement. With the ESPA-29 and with the PARQ/C, most negative outcomes of control are referred to emotional variables (mainly self-esteem), while substance use, problematic behavior and academic achievement usually show no significant association (or even show a protective role of control, in one article). However, this is not the case with other instruments. Especially with the PSI, while most positive outcomes of a high parental control are indeed behavioral ones (lower substance use, better academic achievement, less problematic behavior), there is also improvement in emotional outcomes (self-esteem, psychological wellness).

It seems therefore that, when determining whether parental control is good or bad for adolescent children, the main issue is not the culture or the outcomes assessed, but the kind of control one is measuring. When authors study behavioral control, measured as monitoring and rule setting (with instruments such as the PSI or the EEEP), control appears as unequivocally beneficial. On the contrary, when control is understood as coercion (with the ESPA-29), it appears as unequivocally harmful.

One might then wonder which components of parental control are beneficial or detrimental. The PSI and the EEEP define their behavioral control dimensions as rule setting and monitoring. And these dimensions are always associated with better outcomes, which suggests that these components would be beneficial for the child’s development. These components are also present in the control dimension of the PARQ/C; however, in this instrument, both monitoring and rule setting are expressed as exaggerated. And the effect of this dimension is not so clear. On the contrary, the control dimension of the ESPA is composed of verbal scolding, physical punishment, and revoking privileges. And this dimension is always associated with negative outcomes. Maybe not all the three components are detrimental, but the whole scale seems to be so. Revoking privileges seems coherent with the behavioral control measures, while physical punishment doesn’t. Future studies might analyze each component separately in order to better understand which types of parental control are beneficial for adolescents.

Then, when parental control is mentioned in a study, it is necessary to clarify whether one is referring to rule setting, monitoring, physical punishment or other types of control. And, if we want to compare results from different studies, we must make sure that we are comparing equivalent constructs. Apparently, it is principally known that there are different types of parental control; however, the results of our review show that this is too often not taken into account. The first studies on parenting styles, mainly in the US, found that both parental control and involvement (both present in the authoritative style) were positive for children. If we find, in a different country, that control seems negative, we shouldn’t just say that this country is different. First, we should check whether we are understanding control in the same way (or at least in similar ways). A good inter-country comparison should be done with the same instruments. Actually, a European study [31] expected to find positive correlates of parental control, except in the Southern countries, where the opposite was expected. Actually, the result was that control was harmful in all the studied countries (though not all outcomes achieved significant associations). The conclusion might be that, with the instrument used (PARQ/C), control seems to be detrimental, regardless of the country.

Therefore, attention must be paid to the scales used in constructing parenting styles. According to the pioneers in the study of parenting styles, the authoritative style consisted on high levels of involvement and control. And this control was understood as establishing and maintaining family rules [2,3,87]. Not all measures of ‘control’ can be used to define authoritative parenting.

Our study has some limitations. First, the search was limited to two databases (Web of Science and Scopus), to a specific time window (2000–2017), to two languages (English and Spanish), to some specific keywords, and to some additional criteria. Therefore, we may have missed some relevant studies. However, we believe the search was broad enough, and that the most important articles in the field were probably included. Furthermore, exhaustiveness is not crucial in this study. Had we found more studies, there is no reason to expect that the results would vary in a relevant way. Second, we have not run a meta-analysis, or a quantitative comparison of the different studies. We have just ‘counted’ how many studies draw positive or negative results, disregarding the quality of the study, the sample size, the statistical power, and the effect size. However, this has not been necessary either. All studies using PSI or EEEP yielded neutral or positive outcomes for parental control, while all studies with ESPA-29 yielded neutral or negative results. In this case, there is no need to take the comparative weight of each study into account. In addition, running a meta-analysis would have required to focus on one single outcome, which would have limited this review in different ways. Third, other relevant variables might play an important role in the relationship between parental control and adolescent outcomes. For example, parental control might have a different role for early adolescents than for late adolescents or young adults, or a different role for boys than for girls. However, regarding age, we do not appreciate a relation between ages and results among the studies using one same instrument (see Table 4). Differences by adolescent’s sex or by other variables (e.g., general versus clinical samples) might be studied, but this would exceed our objectives, and most articles did not report differences by sex. Moreover, no article showed a significant association for one sex, and a non-significant or opposed association for the other sex.

One additional concern might be raised regarding the direction of causality. Most studies are cross-sectional, and causality cannot be inferred. For example, the measures of control might just reflect the children’s willingness to obey [88]. Similarly, some measures of monitoring asses the degree of parental knowledge regarding the child’s behavior; this knowledge might also be mediated by the child’s self-disclosure [89,90]. This implies a recommendation to increase longitudinal and (semi-)experimental research, where these issues are more clearly stated.

The study also has notable strengths. First, this study is, to the best of our knowledge, the first one that analyzes the correlates of parental control depending on the instruments used. We believe this was necessary, given the lack of clarity and the confusion existing on this issue. Second, we have found and compared 5 instruments, used in 55 studies, in many different countries from the 5 continents, and studying many different outcomes. Third, we have found clear differences in the results obtained with the different measuring instruments, which suggests that differences don’t seem to be related to cultures, but to the instruments. In order to verify this issue, an empirical study should be done, comparing different instruments within the same country, and/or comparing different countries using the same instrument.

## 5. Conclusions

Parenting styles are consensually defined by two dimensions: closeness and (behavioral) control. When studying parental control, behavioral control (rules, monitoring) and coercion should be distinguished. There used to be a consensus about the benefits of parental control, but this consensus has partially vanished in some regions during the last decades, due to the confusion regarding different ways of measuring control. Some authors use coercion as their definition of control, and they use this dimension in their construction of parenting styles. Furthermore, they conclude that control and authoritativeness are harmful for adolescent development in some countries (in an increasing number of countries). However, evidence keeps showing that, in different world regions and with regard to different outcomes, behavioral control is positive for children.

In summary, this review finds that some instruments show good consequences for parental control, while some others show bad consequences, and that the difference seems to be in the way of measuring control. Parental monitoring and rule setting seem to be beneficial, while coercion seems to be detrimental to adolescents. This gives us a deeper insight in understanding the role of parental control in children development.

The need to test this conclusion more definitely in well-designed studies is suggested for future research. Specifically, in order to compare the constructs measured by different instruments, studies might be run with two instruments on the same sample. Moreover, different aspects of behavioral control might be assessed separately to test possible differential associations with outcomes. Regarding the issue of intercultural differences, studies might be done in different cultures with the same instrument.

## Figures and Tables

**Figure 1 ijerph-16-03157-f001:**
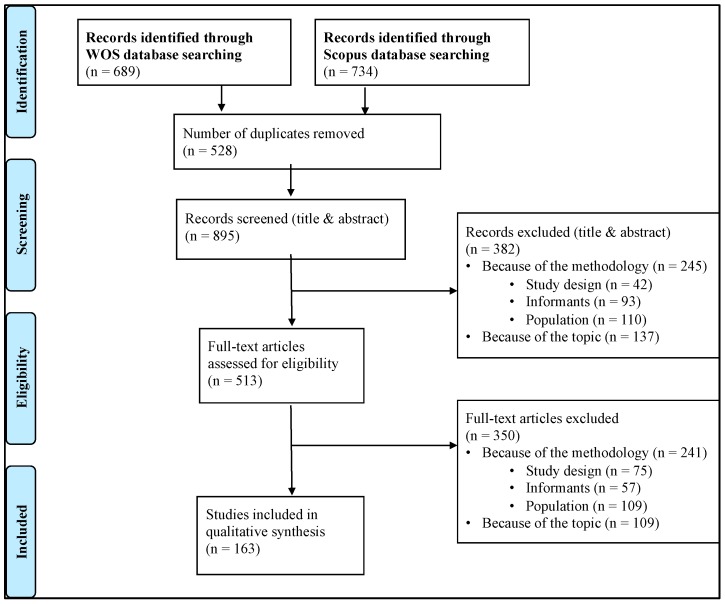
Phases of the systematic review (diagram based on the PRISMA statement [38]).

**Table 1 ijerph-16-03157-t001:** Search criteria in the systematic review.

Field Name in Web of Science ^1^	Field Name in Scopus	Search Criteria ^2^
Title	Article title	“parenting style*” OR “parental style*” OR “socialization style*” OR “parenting practice*” OR “family socialization”
Theme	Title-abs-key	((warmth OR affection OR acceptance OR responsiveness OR support) AND (control OR strictness OR supervision OR monitoring OR coercion)) OR (authoritative OR democratic OR authoritarian OR neglectful OR permissive OR indulgent)

^1^ The Web of Science search was performed with the “all databases” option; ^2^ Items had to fulfill both criteria.

**Table 2 ijerph-16-03157-t002:** Instruments more frequently used in the analyzed studies.

Instrument	No. of Articles	No. of Countries	Which Countries	Dimensions	No. of Items
PSI (Parenting Styles Index)	26	12	Germany, Brazil, Canada, China, Spain, United States, Holland, India, Iran, Iceland, Italy, Nigeria.	Involvement/responsiveness Psychological autonomy-granting Strictness/supervision/demandingness	22/32
ESPA-29 (Parental Socialization Scale)	10	3	Brazil, Spain, United States	Acceptance/involvement Strictness/imposition	29
CRPBI (Child’s Report of Parental Behavior Inventory)	6	2	United States, Spain	Autonomy, autonomy and love, love, love and control, control, control, and hostility, hostility, hostility and autonomy.	260
EEEP (Escala para la Evaluación del Estilo Parental)	5	1	Spain	affection/communication, promotion of autonomy, behavioral control, psychological control, disclosure and humor.	41
PARQ/C (Parental Acceptance-Rejection/Control Questionnaire)	5	6	Slovenia, Spain, United Kingdom, Portugal, Czech Republic, Sweden.	Warmth/affection Hostility/aggression Indifference/neglect Undifferentiated rejection Control	73/29

Note: the control-related dimensions are underlined.

**Table 3 ijerph-16-03157-t003:** Outcomes more frequently analyzed.

Behavioral Outcomes		Emotional Outcomes	
Descriptions	No. ^a^	Descriptions	No. ^a^
Tobacco use Substance abuse Alcohol abuse	17	Self-concept Self-esteem Self-efficacy	16
Prosocial behavior Behavioral problems Troublemaker Antisocial behavior Delinquency Personal disorder Self-control Aggressive behavior Hostility	15	Depression Anxiety Emotional instability Internalizing disorder Psychological strength Negative world perception Psychological adjustment Emotional irresponsibility	9
Academic performance School achievement Have finished secondary school Being a repeater	12	Life satisfaction	4
Bullying involvement	3	Peer attachment Support and closeness to siblings Parent attachment	3
Reaction to the conflict	1	Creativity	1
Early sexual relationships	1	Emotional intelligence	1
Suicide attempts	1	Procrastination	1
Sexist attitude	1	Compassion	1
		Moral reasoning	1

^a^ Number of articles that analyze the corresponding group of outcomes.

**Table 4 ijerph-16-03157-t004:** Detailed associations between control and outcomes.

Instrument and Article			Country/Culture	Times Cited ^a^	Age Range or Mean (M) (Years)	Difference between Outcomes from AV vs. I Styles			Control Dimension Associated with Outcomes			Other ^h^
						AV < I ^b^	ns ^c^	AV > I ^d^	- ^e^	ns ^f^	+ ^g^	
CRPBI	Chassin, 2005	[48]	US	120	10–17		Tobacco use					
	Walker-Barnes, 2004	[49]	US (Hispanics)	31	13–18				Substance use			
	Ozer, 2013	[50]	US (Mexicans)	15	12–15				Depressive symptoms			
	Llorca, 2017	[51]	Spain	4	13–16		Aggressiveness (mother)	Attachment to peers; Self-efficacy; Aggressiveness (father)				
	Tur-Porcar, 2017	[52]	Spain	0	14–18							Internet use
	Carlo, 2011	[53]	Spain	85	9–14				Sympathy; Prosocial behavior (father)		Moral reasoning; Prosocial behavior (mother)	
EEEP	Álvarez-García, 2016	[54]	Spain	11	12–18						School fights; Antisocial behavior; Negative social relations	
	Gómez-Ortiz, 2014	[55]	Spain	24	12–18					Bullying		
	Gómez-Ortiz, 2015	[42]	Spain	10	12–18							Bullying Resilience Attachment Family awareness Child trauma
	Oliva, 2007	[56]	Spain	3	12–17				Internalizing problems (mother)	Internalizing problems (father); Externalizing problems (mother); Life satisfaction (mother)	Substance use; Positive development; Externalizing problems (father); Life satisfaction (father)	
	Oliva, 2008	[43]	Spain	48	12–17							Adjustment Life satisfaction Self-esteem
PSI	Donath, 2014	[17]	Germany	37 *	M = 15.3			Suicide attempt			Suicide attempt	
	Paiva, 2012	[57]	Brazil	6	14–19		Alcohol use					
	Tondowski, 2015	[22]	Brazil	7 *	13–18			Tobacco use				
	Valente, 2017	[58]	Brazil	13	11–15			Substance use				
	Garg, 2005	[59]	Canada India	24	13–15			Academic achievement				
	Chao, 2001	[60]	China US	377	14–18		Academic achievement					
	Miller, 2002	[61]	US	23 *	11–14			Negative reaction to conflict				
	Pittman, 2001	[62]	US	71	15–18		Internalizing disorder; Academic achievement	Externalizing disorder				
	Bahr, 2010	[63]	US	48	12–18		Alcohol use	Alcohol use				
	Milevsky, 2011	[64]	US	15	14–17			Support to siblings; Closeness to siblings				
	Milevsky, 2007	[65]	US	179 *	14–17		Self-esteem and Life satisfaction (mother) Depression	Self-esteem and Life satisfaction (father)				
	Milevsky, 2008	[66]	US	19 *	14–17			Psychological well-being				
	Osorio, 2016	[8]	Spain	8	13–17		Self-esteem	Academic achievement		Self-esteem	Academic achievement	
	Parra, 2006	[6]	Spain	40	13–17						Substance use; Externalizing problems	
	Den Exter Blokland, 2001	[67]	Netherlands	9	13–17		Delinquency; Self-control					
	Huver, 2007	[68]	Netherlands	21	M = 15.35						Tobacco use	
	Abdi, 2015	[69]	Iran	2	14–17			Life satisfaction; Academic achievement; Psychological strength				
	Mozayyeni, 2017	[70]	Iran	0	13–17			Self-esteem; Procrastination				
	Dehyadegary, 2012	[71]	Iran	0 *	15–18			Academic achievement				
	Mehrinejad, 2015	[72]	Iran	3	14–17			Creativity				
	Pour, 2015	[73]	Iran	2	13–15			Academic achievement				
	Eshrati, 2017	[74]	Iran	0 *	M = 17			Behavioral problems				
	Adalbjarnardottir, 2001	[75]	Iceland	91	14–17			Substance use				
	Blondal, 2009	[76]	Iceland	34	14–21			Academic achievement				
	Moscatelli, 2011	[5]	Italy	2	16–18		Self-efficacy	Self-esteem		Self-esteem	Self-efficacy	
	Adekeye, 2015	[77]	Nigeria	0	15–19			Emotional Intelligence				
ESPA-29	Martínez, 2007	[27]	Brazil	47	11–15	Self-esteem	Self-esteem					
	Martínez, 2017	[78]	US	16	14–18				Self-esteem			
	Cerezo, 2015	[79]	Spain	25	9–18		Victim of bulling					
	Fuentes, 2015	[28]	Spain	40	12–17	Self-esteem; Hostility/aggression; Emotional Instability; Negative view of the world	Self-efficacy: Emotional irresponsiveness					
	Gallarin, 2012	[80]	Spain	29	16–19				Attachment; Aggressiveness (mother)	Aggressiveness (father)		
	Garaigordobil, 2012	[81]	Spain	28	11–17		Sexist attitude					
	Gracia, 2012	[82]	Spain	32	12–17		Hostility/aggression; Self-esteem; Emotional irresponsiveness; Emotional instability; Negative view of the world; Academic achievement; Disruptive behavior; Delinquency; Substance use					
	Martínez, 2007	[83]	Spain	63	13–16	Self-esteem						
	Martínez, 2013	[84]	Spain	23	14–17		Disruptive school behavior; Delinquency; Substance use					
	Musitu, 2004	[44]	Spain	101	14–17	Self-esteem	Self-esteem					
PARQ/C	Martinez-Loredo, 2016	[85]	Spain	6	12–16		Alcohol use					
	García, 2009	[29]	Spain	136 *	12–17	Self-esteem; emotional irresponsiveness; Negative view of the world; Academic achievement	Self-esteem; Hostility/aggression Self-concept; Social competence; Academic achievement; Disruptive behavior; Delinquency; Drug use					
	García, 2010	[30]	Spain	66 *	10–14	Self-esteem; Aggressiveness; Emotional instability; Negative view of the world	Self-esteem; Self-efficacy; Emotional irresponsiveness; Academic achievement; Substance use; Disruptive behavior; Delinquency					
	Cablova, 2016	[86]	Czech Republic	0	10–18						Alcohol use	
	Calafat, 2014	[31]	Sweden, United Kingdom, Spain, Portugal, Slovenia, Czech Republic	90 *	11–19	Self-esteem; Academic achievement	Substance use; Personal problems					

^a^ Times cited according to Web of Science, except when an asterisk (*) appears after the number. The asterisk indicates that the article was not found in Web of Science, and the number of times cited according to Scopus is given. ^b^ Outcomes where the Authoritative style scores significantly worse than the Indulgent style. ^c^ Outcomes with no significant differences between the scores of the Authoritative and the Indulgent style. ^d^ Outcomes where the Authoritative style scores significantly better than the Indulgent style. ^e^ Outcomes where Control is significantly associated with poorer results. ^f^ Outcomes not significantly associated with Control, or where positive and negative associations are balanced. ^g^ Outcomes where Control is significantly associated with better results. ^h^ This column means that the article run analyses where the role of the Control dimension could not be isolated from other variables.

**Table 5 ijerph-16-03157-t005:** Results, depending on the instrument used.

	Difference between Outcomes from AV vs. I Styles			Control Dimension Associated with Outcomes			Other Analyses ^g^	Total
Instrument	AV < I ^a^	ns ^b^	AV > I ^c^	- ^d^	ns ^e^	+ ^f^		
PSI		3	21			5		26 ^h^
EEEP					1	2	2	5
CRPBI		1	1	3			1	6
PARQ/C	3	1				1		5
ESPA29	4	4		2				10

^a^ Number of articles where the Authoritative style obtains significantly worse outcomes than the Indulgent style. ^b^ Number of articles where there are no significant differences between the outcomes of the Authoritative and the Indulgent style. ^c^ Number of articles where the Authoritative style obtains significantly better outcomes than the Indulgent style. ^d^ Number of articles where Control is significantly associated with poorer outcomes. ^e^ Number of articles where Control is not significantly associated with outcomes, or where positive and negative outcomes are balanced. ^f^ Number of articles where Control is significantly associated with better outcomes. ^g^ Number of articles with analyses where the role of the Control dimension could not be isolated from other dimensions. ^h^ This total differs from the sum of numbers in this row (32) because 3 articles use both categorical (styles) and dimensional analyses.

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
