# Peer review of "Measurement and Function of the Control Dimension in Parenting Styles: A Systematic Review"

_ijerph, 2019, doi:10.3390/ijerph16173157_

Round 1

Reviewer 1 Report

Dear authors,

I think the article has improved to be considered for publication. Many thanks for your trust.

Best regards.

Author Response

Reviewer 1

Dear authors,

I think the article has improved to be considered for publication. Many thanks for your trust.

Best regards.

We appreciate this comment.

Reviewer 2 Report

This revised version of this article has improved considerably from the first one I revised.

It is now better understood, that parenting psychological and behavioral control, be negatively and positively related to the affection dimension. Control dimension, on the other hand, should by theoretically and empirically orthogonal respect to affection dimension (e.g., Steinberg, 2005). I think that authors should to explain better the relationship between parental practices and main parental dimensions, especially when you discuss your results. In order to understand what you have written, the molar parts and the parental practices are the same concept (e.g., Darling & Steinberg, 1993, p. 489).

References

Darling, N., & Steinberg, L. (1993). Parenting style as context: An integrative model. Psychological Bulletin, 113, 487-496. doi:10.1037/0033-2909.113.3.487

Steinberg, L. (2005). Psychological control: Style or substance? In J. G. Smetana (Ed.), New directions for child and adolescent development: Changes in parental authority during adolescence (pp. 71-78). San Francisco: Jossey-Bass. doi:10.1002/cd.129

Author Response

Reviewer 2

-This revised version of this article has improved considerably from the first one I revised.

We appreciate this comment.

-It is now better understood, that parenting psychological and behavioral control, be negatively and positively related to the affection dimension. Control dimension, on the other hand, should by theoretically and empirically orthogonal respect to affection dimension (e.g., Steinberg, 2005).

We appreciate this comment, too.

-I think that authors should to explain better the relationship between parental practices and main parental dimensions, especially when you discuss your results. In order to understand what you have written, the molar parts and the parental practices are the same concept (e.g., Darling & Steinberg, 1993, p. 489).

We added some comments in the Discussion, regarding the specific components of the control dimensions as measured by the different  instruments. We believe we cannot go further, since the articles analyzed did not report results related to the different components.                           

One might then wonder which components of parental control are beneficial or detrimental. The PSI and the EEEP define their behavioral control dimensions as rule setting and monitoring. And these dimensions are always associated with better outcomes, which suggests that these components would be beneficial for the child’s development. These components are also present in the control dimension of the PARQ/C; however, in this instrument, both monitoring and rule setting are expressed as exaggerated. And the effect of this dimension is not so clear. On the contrary, the control dimension of the ESPA is composed of verbal scolding, physical punishment, and revoking privileges. And this dimension is always associated with negative outcomes. Maybe not all the three components are detrimental, but the whole scale seems to be so. Revoking privileges seems coherent with the behavioral control measures, while physical punishment doesn’t. Future studies might analyze each component separately in order to better understand which types of parental control are beneficial for children.

Reviewer 3 Report

This manuscript address the issue of parental control practice. By reviewing the literature, authors indicate that different types of control relate differently to the children adjustment or maladjustment. Mostly several control practices labels (e.g., psychological control, firm control or lax control) were write by Schaefer (1965a) who is also the author of the CRPBI questionnaire (Schaefer, 1965b) (e.g., table 2 of this manuscript).

Literature knows that psychological control is negatively related to child adjustment and behavioral control positively. Psychological control implies control and rejection (opposite to love/affection, but in same dimension), while behavioral control implies control but with love/affection. When researchers investigate parenting styles (authoritarian, authoritative, indulgent, or neglectful) dimensions, this axe dimensions should be orthogonal and not collinear (e.g., Darling & Steinberg, 1993, pp. 491-492; Smetana, 1995, p. 299; Steinberg, 2005, p. 71). Psychological control, high control and low love/affection (i.e., rejection), is an authoritarian characteristic; otherwise, behavioral control (high control and low love/affection) is an authoritative characteristic.

It is a methodological error capture the axis of control, which has later been given other names (e.g., Steinberg, 2005), both with specifically psychological control or with specifically behavioral control parenting practices. However, could be possible adequately measure control with subscales of psychological control and behavioral control whether this composed control measure be orthogonal to love/affection. It should be verified that the factorial structure of the two axes be orthogonal (control respect to affection), on the other hand, also that psychological control is negatively related to affection and behavioral control positively to affection. It is true that initial Baumrind's "Y" model does not allow you to analyze the four styles as theoretically defined Maccoby and Martin (1983). You write that“(Parental Authority Questionnaire) [37,38] does not have any specific scale of parental control or authority” (lin. 163-164). PAQ questionnaire have exactly three, authoritarian, authoritative and permissive ‘‘control’’ dimensions, as initially proposed Baumrind's "Y" model to define this three parenting typologies (Baumrind, 1967, 1971). Its well know that as Lamborn, Mounts, Steinberg, and Dornbusch (1991) noted "most discussions and empirial tests of Baumrind's model. . . ignore variations in warmth among families characterized by low levels of control, grouping these families together into a single category labeled "permissive" (p. 1050). However still you noted, year 2019, that Parental Authority Questionnaire is one of instruments used most frequently. When it is well known, that Baumrind's "Y" model did not differentiate indulgent from neglectful parenting styles.

On the other hand, it is true that the acceptance/involvement axe of the PSY questionnaire capture love/affection. Otherwise, is not true that psychological autonomy granting was used for measured the four parenting styles (e.g., Lamborn, Mounts, Steinberg, & Dornbusch, 1991, p. 1053): “The psychological autonomy dimension appears to be important in defining authoritativeness but less so in differentiating among authoritative, authoritarian, indulgent, and neglectful families”. Additionally, the strictness/supervision items to capture control (“In a typical week, what is the latest you can stay out on SCHOOL NIGHTS”, or, “How much do your parents TRY to know”) were hard criticized, for example, because cannot distinguish authoritarian from authoritative monitoring (e.g., Calafat, García, Juan, Becoña, & Fernández-Hermida, 2014; Kerr & Stattin, 2000; Stattin & Kerr, 2000).

The ESPA29 questionnaire (Moreno-Ruiz, Estévez, Jiménez, & Murgui, 2018; Garcia, Serra, Garcia, Martinez, & Cruise, 2019) measures parental strictness/supervision (also, as you said, so called control), with three molar components or parental practices measures: Verbal scolding, “He/she scolds me”, physical punishment “He/she hits me”, and revoking privileges, “He/she takes something away from me”. Three molar components are specifically related to when child behaves inappropriately. There are evidence that ESPA29 two main axis are orthogonal and have a coherent main axis composition from parenting practices, or molar, sub-scales (e.g., Martinez, Garcia, Fuentes, Veiga, Garcia, Rodrigues, Cruise, & Serra, 2019).

The PARQ/C (Parental Acceptance-Rejection/Control Questionnaire) measure, as you said, five   molar components or parental practices measures: warmth/affection, hostility/aggression, Indifference/neglect, undifferentiated rejection, and control. In this case, warmth/affection and control have usually used to distinguish the four parenting styles (e.g., Fuentes, García-Ros, Pérez-González, & Sancerni, 2019; Garcia & Gracia, 2009, 2010). In addition, there are evidence that these two dimensions are orthogonal (e.g., Garcia, Serra, Zacares, & Garcia, 2018).

In short, the question is how parent control and affection (as you said) influence the adjustment of the children. On the one hand, is true that is need measure de factorial structural invariance between different parental practices along different countries, contexts, environments, demographic variables, etc. This is the first requisite for reproducibility, replicability, and generalizability (e.g., Stanley, Carter, & Doucouliagos, 2018), but also discuss whether theoretical concepts are included in measures. Four parenting styles model defined the same control dimension for authoritarian and authoritative parenting style and the same love dimension for authoritative and indulgent parenting style. As you find, recently research noted that indulgent parenting style (warmth but non-control) is similar or even better than authoritative parenting (warmth and control).

Your manuscript should clarify the relationship between practices and styles as discussed in the literature. The work is of interest because it reflects the most important discussions and developments in this research field. However, authors have only considers the measures and it lacks that at least it comments minimally on the theoretical background that have appeared throughout 50 years.

Baumrind, D. (1967). Child cares practices anteceding three patterns of preschool behavior. Genetic Psychology Monographs, 75, 43-88.

Baumrind, D. (1971). Current patterns of parental authority. Developmental Psychology, 4(1, Pt.2), 1-103. doi:10.1037/h0030372

Calafat, A., García, F., Juan, M., Becoña, E., & Fernández-Hermida, J. R. (2014). Which parenting style is more protective against adolescent substance use? Evidence within the European context. Drug and Alcohol Dependence, 138, 185-192. doi:10.1016/j.drugalcdep.2014.02.705

Darling, N., & Steinberg, L. (1993). Parenting style as context: An integrative model. Psychological Bulletin, 113, 487-496. doi:10.1037/0033-2909.113.3.487

Fuentes, M. C., García-Ros, R., Pérez-González, F., & Sancerni, D. (2019). Effects of parenting styles on self-regulated learning and academic stress in Spanish adolescents. International Journal of Environmental Research and Public Health, 16(2778), 1-19. doi:10.3390/ijerph16152778

García, F., & Gracia, E. (2009). Is always authoritative the optimum parenting style? Evidence from Spanish families. Adolescence, 44(173), 101-131.

García, F., & Gracia, E. (2010). What is the optimum parental socialisation style in Spain? A study with children and adolescents aged 10-14 years [¿Qué estilo de socialización parental es el idóneo en España? Un estudio con niños y adolescentes de 10 a 14 años]. Infancia y Aprendizaje, 33, 365-384. doi:10.1174/021037010792215118

Garcia, F., Serra, E., Garcia, O. F., Martinez, I., & Cruise, E. (2019). A third emerging stage for the current digital society? Optimal parenting styles in Spain, the United States, Germany, and Brazil. International Journal of Environmental Research and Public Health, 16(2333), 1-20. doi:10.3390/ijerph16132333

Lamborn, S. D., Mounts, N. S., Steinberg, L., & Dornbusch, S. M. (1991). Patterns of competence and adjustment among adolescents from authoritative, authoritarian, indulgent, and neglectful families. Child Development, 62, 1049-1065. doi:10.1111/j.1467-8624.1991.tb01588.x

Martinez, I., Garcia, F., Fuentes, M. C., Veiga, F., Garcia, O. F., Rodrigues, Y., Cruise, E., & Serra, E. (2019). Researching parental socialization styles across three cultural contexts: Scale ESPA29 bi-dimensional validity in Spain, Portugal and Brazil. International Journal of Environmental Research and Public Health, 16(197), 1-14. doi:10.3390/ijerph16020197

Moreno-Ruiz, D., Estévez, E., Jiménez, T. I., & Murgui, S. (2018). Parenting style and reactive and proactive adolescent violence: Evidence from Spain. International Journal of Environmental Research and Public Health, 15(2634), 1-13. doi:10.3390/ijerph15122634

Schaefer, E. S. (1965). A configurational analysis of children's reports of parent behavior. Journal of Consulting Psychology, 29, 552-557. doi:10.1037/h0022702

Schaefer, E.S. (1959). A circumplex model for maternal behavior. Journal of Abnormal and Social Psychology59, 226-235.

Schaefer, E.S. (1961). Multivariate measurement and factorial structure of children’s perceptions of maternal and paternal behavior. American Psychologist, 16, 345-346 (abstract).

Schaefer, E.S. (1965a). Children’s Reports of Parental Behavior: An inventory. Child Development, 36, 413-424.

Smetana, J. G. (1995). Parenting styles and conceptions of parental authorityduring adolescence. Child Development, 66, 299-316.

Stanley, T. D., Carter, E. C., & Doucouliagos, H. (2018). What meta-analyses reveal about the replicability of psychological research. Psychological Bulletin, 144, 1325-1346.

Stattin, H., & Kerr, M. (2000). Parental monitoring: A reinterpretation. Child Development, 71, 1072-1085. doi:10.1111/1467-8624.00210

Steinberg, L. (2005). Psychological control: Style or substance? In J. G. Smetana (Ed.), New directions for child and adolescent development: Changes in parental authority during adolescence (pp. 71-78). San Francisco: Jossey-Bass. doi:10.1002/cd.129

Author Response

Reviewer 3

Comments: This manuscript address the issue of parental control practice. By reviewing the literature, authors indicate that different types of control relate differently to the children adjustment or maladjustment. Mostly several control practices labels (e.g., psychological control, firm control or lax control) were write by Schaefer (1965a) who is also the author of the CRPBI questionnaire (Schaefer, 1965b) (e.g., table 2 of this manuscript).

Literature knows that psychological control is negatively related to child adjustment and behavioral control positively. Psychological control implies control and rejection (opposite to love/affection, but in same dimension), while behavioral control implies control but with love/affection. When researchers investigate parenting styles (authoritarian, authoritative, indulgent, or neglectful) dimensions, this axe dimensions should be orthogonal and not collinear (e.g., Darling & Steinberg, 1993, pp. 491-492; Smetana, 1995, p. 299; Steinberg, 2005, p. 71). Psychological control, high control and low love/affection (i.e., rejection), is an authoritarian characteristic; otherwise, behavioral control (high control and low love/affection) is an authoritative characteristic.

Reply: If we understand correctly, you are defining “psychological control” as “high control and low love/affection”; and “behavioral control” as “high control and low love/affection”. This is of course a possible way of defining these constructs, but this is not what we found in the instruments assessed. The instruments that use these terms (“psychological control” and “behavioral control”) understand them as dimensions that are different from the love/affection dimension. Of course, it can be discussed whether the dimensions are actually independent (orthogonal), but this is a different issue.

Comments: It is a methodological error capture the axis of control, which has later been given other names (e.g., Steinberg, 2005), both with specifically psychological control or with specifically behavioral control parenting practices. However, could be possible adequately measure control with subscales of psychological control and behavioral control whether this composed control measure be orthogonal to love/affection. It should be verified that the factorial structure of the two axes be orthogonal (control respect to affection), on the other hand, also that psychological control is negatively related to affection and behavioral control positively to affection. It is true that initial Baumrind's "Y" model does not allow you to analyze the four styles as theoretically defined Maccoby and Martin (1983).

Reply: It is not our goal to determine which instruments are more adequate, or which ones capture the essence of the “control” dimension. Since we are following the model proposed by Baumrind (1967) and Maccoby and Martin (1983), we include their definitions of “parental control” and of “authoritative parenting” as a key to guide our search for control dimensions in the instruments analyzed. However, we have included instruments with different definitions of this dimension.

Another article might discuss which of the existing instruments are more adequate, both in terms of the theoretical definition and of the empirical functioning (for example, regarding orthogonality). But our review cannot afford that scope except by becoming too long. Furthermore, many studies don’t report data on orthogonality.

Comments: You write that“(Parental Authority Questionnaire) [37,38] does not have any specific scale of parental control or authority” (lin. 163-164). PAQ questionnaire have exactly three, authoritarian, authoritative and permissive ‘‘control’’ dimensions, as initially proposed Baumrind's "Y" model to define this three parenting typologies (Baumrind, 1967, 1971). Its well know that as Lamborn, Mounts, Steinberg, and Dornbusch (1991) noted "most discussions and empirial tests of Baumrind's model. . . ignore variations in warmth among families characterized by low levels of control, grouping these families together into a single category labeled "permissive" (p. 1050). However still you noted, year 2019, that Parental Authority Questionnaire is one of instruments used most frequently. When it is well known, that Baumrind's "Y" model did not differentiate indulgent from neglectful parenting styles.

Reply: The PAQ dimensions are not “pure” control dimensions, since they include control and love/affection components. Parents who score high in authoritarianism, and those who score high in authoritativeness, show high degrees of control. Therefore we cannot isolate the control dimension. This may have advantages and disadvantages, but our analyses cannot deal with that kind of instrument.

Comments: On the other hand, it is true that the acceptance/involvement axe of the PSY questionnaire capture love/affection. Otherwise, is not true that psychological autonomy granting was used for measured the four parenting styles (e.g., Lamborn, Mounts, Steinberg, & Dornbusch, 1991, p. 1053): “The psychological autonomy dimension appears to be important in defining authoritativeness but less so in differentiating among authoritative, authoritarian, indulgent, and neglectful families”.

Reply: We agree with this. As we say (lines 191-192): “when parenting styles are built as categories, the scales used are involvement and behavioral control (not psychological control)”.

Comments: Additionally, the strictness/supervision items to capture control (“In a typical week, what is the latest you can stay out on SCHOOL NIGHTS”, or, “How much do your parents TRY to know”) were hard criticized, for example, because cannot distinguish authoritarian from authoritative monitoring (e.g., Calafat, García, Juan, Becoña, & Fernández-Hermida, 2014; Kerr & Stattin, 2000; Stattin & Kerr, 2000).

Reply: We are aware of these criticisms. Maybe an article should address this issue. However we are not trying to compare two kinds of monitoring (authoritarian VS. authoritative). We are exposing different publications apparently based on Baumrind and Maccoby & Martin model of parenting styles, where authors use different types of control dimensions or where they use different control dimensions to build parenting styles following orthogonality criteria (as they claim in their publications).

We are not trying to judge the questionnaires. We are just showing different articles and different ways to understand and use control dimension and parenting styles and how they relate to outcomes.

Comments: The ESPA29 questionnaire (Moreno-Ruiz, Estévez, Jiménez, & Murgui, 2018; Garcia, Serra, Garcia, Martinez, & Cruise, 2019) measures parental strictness/supervision (also, as you said, so called control), with three molar components or parental practices measures: Verbal scolding, “He/she scolds me”, physical punishment “He/she hits me”, and revoking privileges, “He/she takes something away from me”. Three molar components are specifically related to when child behaves inappropriately. There are evidence that ESPA29 two main axis are orthogonal and have a coherent main axis composition from parenting practices, or molar, sub-scales (e.g., Martinez, Garcia, Fuentes, Veiga, Garcia, Rodrigues, Cruise, & Serra, 2019).

The PARQ/C (Parental Acceptance-Rejection/Control Questionnaire) measure, as you said, five   molar components or parental practices measures: warmth/affection, hostility/aggression, Indifference/neglect, undifferentiated rejection, and control. In this case, warmth/affection and control have usually used to distinguish the four parenting styles (e.g., Fuentes, García-Ros, Pérez-González, & Sancerni, 2019; Garcia & Gracia, 2009, 2010). In addition, there are evidence that these two dimensions are orthogonal (e.g., Garcia, Serra, Zacares, & Garcia, 2018).

Reply: As above mentioned, we are not judging or analyzing how ESPA-29, PARQ/C. In any case, we have added a sentence regarding orthogonality when commenting these instruments (lines 209-211).

Comments: In short, the question is how parent control and affection (as you said) influence the adjustment of the children. On the one hand, is true that is need measure de factorial structural invariance between different parental practices along different countries, contexts, environments, demographic variables, etc. This is the first requisite for reproducibility, replicability, and generalizability (e.g., Stanley, Carter, & Doucouliagos, 2018), but also discuss whether theoretical concepts are included in measures. Four parenting styles model defined the same control dimension for authoritarian and authoritative parenting style and the same love dimension for authoritative and indulgent parenting style. As you find, recently research noted that indulgent parenting style (warmth but non-control) is similar or even better than authoritative parenting (warmth and control).

Reply: As we find, the indulgent parenting style is similar or better when assessed by certain instruments (PARQ/C or ESPA-29), but not when assessed with other ones (PSI or EEEP).

Comments: Your manuscript should clarify the relationship between practices and styles as discussed in the literature. The work is of interest because it reflects the most important discussions and developments in this research field. However, authors have only considers the measures and it lacks that at least it comments minimally on the theoretical background that have appeared throughout 50 years.

Reply: We added some comments in the Discussion, regarding the specific components of the control dimensions as measured by the different  instruments. We believe we cannot go further, since the articles analyzed did not report results related to the different components.

One might then wonder which components of parental control are beneficial or detrimental. The PSI and the EEEP define their behavioral control dimensions as rule setting and monitoring. And these dimensions are always associated with better outcomes, which suggests that these components would be beneficial for the child’s development. These components are also present in the control dimension of the PARQ/C; however, in this instrument, both monitoring and rule setting are expressed as exaggerated. And the effect of this dimension is not so clear. On the contrary, the control dimension of the ESPA is composed of verbal scolding, physical punishment, and revoking privileges. And this dimension is always associated with negative outcomes. Maybe not all the three components are detrimental, but the whole scale seems to be so. Revoking privileges seems coherent with the behavioral control measures, while physical punishment doesn’t. Future studies might analyze each component separately in order to better understand which types of parental control are beneficial for children.

In short, this discussion is a very interesting topic. It might become an article on its own. Instruments have pros and cons, and some of them have appeared here. But we believe we cannot (and must not) face all these issues in our article. The article does not aim to compare instruments, trying to show which one is better or worse. We describe each control dimension, and report how it is related to adolescent outcomes, suggesting that different instruments lead to different results. A future article might confront all these instruments, showing pros and cons, including the definition of the constructs, orthogonality of dimensions and other topics. But this goes beyond the goals of the present article.

Reviewer 4 Report

This is an interesting paper potentially adding to previous literature. I think it can be considered for publication, should the authors be prepared to address some points, as follows:

- Given the special focus on adolescents, the authors should add at least one paragraph in the introduction to discuss the specificity of this developmental phase and the possible specificity of parenting practices with adolescents.

- In this line, I suggest adding reference to adolescence in the title.

- The authors correctly use the PRSIMA model for systematic reviews. It would be useful to know if and how the included articles impacted on the field, by checking the number of citations for each paper (at least for the 163 papers included in the synthesis).

- It seems that the authors did not focus on the possible differences and outcomes of control dimension and parenting in general in clinical/at risk or normative samples (parents and/or adolescents with psychopathological risk, for example). I suggest adding a brief passage and related references about this rooting on specific literature (Cerniglia, Cimino, Tafà, Marzilli, Ballarotto, & Bracaglia, 2017; Tafà, Cimino, Ballarotto, Bracaglia, Bottone, Cerniglia, 2017).

- Correctly, the authors cited the most frequent outcomes of parental style. However, less attention has been given to the predictors of parental styles (again, parental psychopathological risk; poverty; etc.). I suggest adding a brief passage on this.

Line 42: a citation is missing

Lines 49-55: the citation should show page numbers also, as it is a literal citation.

I suggest an English editing. The quality of language is fair, but in some parts the manuscript is difficult to follow.

Author Response

Reviewer 4

This is an interesting paper potentially adding to previous literature.

We appreciate this comment.

I think it can be considered for publication, should the authors be prepared to address some points, as follows:

- Given the special focus on adolescents, the authors should add at least one paragraph in the introduction to discuss the specificity of this developmental phase and the possible specificity of parenting practices with adolescents.

We have added a paragraph at the beginning of the Introduction.

- In this line, I suggest adding reference to adolescence in the title.

We might use this title, if the Editor agrees.

Measurement and Function of the Control Dimension in Parenting Styles with adolescent children: A Systematic Review.

- The authors correctly use the PRSIMA model for systematic reviews. It would be useful to know if and how the included articles impacted on the field, by checking the number of citations for each paper (at least for the 163 papers included in the synthesis).

We have added a column in Table S1 (Annex), indicating this information.

- It seems that the authors did not focus on the possible differences and outcomes of control dimension and parenting in general in clinical/at risk or normative samples (parents and/or adolescents with psychopathological risk, for example). I suggest adding a brief passage and related references about this rooting on specific literature (Cerniglia, Cimino, Tafà, Marzilli, Ballarotto, & Bracaglia, 2017; Tafà, Cimino, Ballarotto, Bracaglia, Bottone, Cerniglia, 2017).

Unfortunately, we did not focus on the possible differences between clinical and normative samples because most articles do not make this distinction. We have added a reference in the Limitations section (line 390).

- Correctly, the authors cited the most frequent outcomes of parental style. However, less attention has been given to the predictors of parental styles (again, parental psychopathological risk; poverty; etc.). I suggest adding a brief passage on this.

There are certainly some factors that could encourage or reduce the possibility of using specific parental styles, for example personal aspects of children (Harris, 2002), family structure (Oliva, Parra, & Arranz, 2008), family size (Coloma, 1993) or socioeconomic level (Steinberg & Silk, 2002). However, our article intentionally focuses on outcomes, since this is the goal of the study. If the Editor believes this is useful, we can add a comment in the Introduction.

Coloma, J. (1993). La familia como ámbito de socialización de los hijos. In J. M. Quintana Cabanas (Ed.), Pedagogía familiar (pp. 31–44). Madrid, España: Narcea Ediciones.

Harris, J. R. (2002). Beyond the nurture assumption: Testing hypotheses about the child’s environment. In J. G. Borkowski, S. L. Ramey, & M. Bristol-Power (Eds.), Monographs in parenting. Parenting and the child’s world: Influences on academic, intellectual, and social-emotional development (pp. 3–20). Mahwah, NJ, US: Lawrence Erlbaum Associates Publishers.

Oliva, A., Parra, Á., & Arranz, E. (2008). Estilos relacionales parentales y ajuste adolescente [Parenting styles and adolescent adjustment]. Infancia y Aprendizaje, 31(1), 93–106. https://doi.org/10.1174/021037008783487093

Steinberg, L., & Silk, J. (2002). Parenting adolescents. In M. H. Bornstein (Ed.), Handbook of parenting, 1 (pp. 103–133). Mahwah, NJ, US: Lawrence Erlbaum Associates Publishers.

-Line 42: a citation is missing

References have been added.

-Lines 49-55: the citation should show page numbers also, as it is a literal citation.

Thanks for noticing this. We added page numbers.

-I suggest an English editing. The quality of language is fair, but in some parts the manuscript is difficult to follow.

We have revised English language.

Round 2

Reviewer 3 Report

Thank you for your review. 

Author Response

Thanks.

This manuscript is a resubmission of an earlier submission. The following is a list of the peer review reports and author responses from that submission.

Round 1

Reviewer 1 Report

I very much appreciate the opportunity to review the paper titled “Measurement and Function of the Control Dimension in Parenting Styles: A Systematic Review”.

This study addresses an important topic, the parenting styles and its main dimensions with a syctematic review. Specifically, authors analyze the discussion about control dimension with controversial and challenging proposals. It seems an interesting article which makes a sound contribution to this traditional debate about family socialization styles. In order to figure this out, the authors used a systematic review of the recent literature (WOS and Scopus, 2000-2017). The paper is well written and collects significant scientific literature that have cover similar topics in relevant databases.

The weakest part of this paper is the search criteria and some essential concepts and how this study can contribute to existing theory and intervention. What is the importance of this manuscript?

Below the authors may find as follows a few specific comments relating to each of the sections:

1.    The authors wrote: “Line 65: The explanation of this phenomenon might not be in the culture, but perhaps in the way of measuring the ‘control’ dimension in the instrument used. In order to know whether the type of instrument is having an impact on the study’s results, it would be good to carry out that same research but using another questionnaire that measures control as supervision and monitoring and not as restriction and coercion."

"Line 317: It seems therefore that, when determining whether parental control is good or bad for children, the main issue is not the culture or the outcomes assessed, but the kind of control one is measuring."

In this paper the concepts of theoretical construct and measures are mixed together. The two concepts are confused by the authors. Obviously, the instruments are conceptual and have an intrinsic meaning. Do you think that parenting measures are different between cultures or do you think the parenting styles are different?

2.    It would be interesting to see more detail around parental warmth. Here, more information is required. Authors should explain clearly the concept of warmth. Others author's thinking that ‘love/affection’ or warmth is a powerful tool and it could be the key of control in indulgent parenting (Moreno-Ruiz, Estévez, Jiménez, & Murgui, 2018). In addition, when you think about parenting styles, it's important to note that have two fundamental dimensions. Should take both types of dimensions to discuss about authoritative and indulgent parenting styles. Do you think that ‘love/affection’ and ‘control/supervision’ are are orthogonal to each other?

3.    Why was the sex of the adolecents not taken into account separately in the sistematic review? The authors indicate “Line 359: Differences by sex or by other variables might be studied, but this would exceed our objectives, and most articles did not report differences by sex”. Some studies have demonstrated a significant difference by sex in the outcomes (Martínez, Murgui, García, & García, 2019; Moreno-Ruiz et al., 2018). I think it is not an aspect previously mentioned in the introduction, but this explanation deserves to be improved and explained.

4.    I believe the authors should review the search criteria in the systematic review. I have searched following some search criteria (Table 1. Search Criteria: (“parenting style*” OR “parental style*” OR “socialization style*” OR “parenting practice*” OR “family socialization”)) and I have obtained other outcomes. For example:

WOK (941 results): Indexes=SSCI Timespan=2000-2017

SCOPUS (1104 results): ( TITLE ( "parenting style" )  OR  TITLE ( "parental style" )  OR  TITLE ( "socialization style" )  OR  TITLE ( "parenting practice" )  OR  TITLE ( "family socialization" ) )  AND  DOCTYPE ( ar )  AND  PUBYEAR  >  2000  AND  PUBYEAR  <  2017  AND  DOCTYPE ( ar )  AND PUBYEAR  >  2000  AND  PUBYEAR  <  2017

5.    Authors should be more specific about the practical implications of their study. Any tips for future research on parenting?

6.    The authors forgot the doi in the references.

I hope that the authors find these comments useful and that they improve the manuscript.

References of review

Martínez, I., Murgui, S., García, O. F., & García, F. (2019). Parenting in the digital era: Protective and risk parenting styles for traditional bullying and cyberbullying victimization. Computers in Human Behavior, 90, 84-92.

Moreno-Ruiz, D., Estévez, E., Jiménez, T., & Murgui, S. (2018). Parenting style and reactive and proactive adolescent violence: Evidence from spain. International journal of environmental research and public health, 15(12), 2634.

Reviewer 2 Report

Present study aims to systematic reviewing the control dimension, measurement and function, in parenting literature. Main manuscript argument is about that one parenting control measures (i.e., measuring behavioral control) are related with children well-adjustment but other parenting control measures (i.e., measuring psychological control) are related with children maladjustment. Authors systematic review analysis have consisted in count the number of studies where different measures give positive or negative relations with control. Study findings indicate that behavioral control measures are well related to children adjustment and psychological control are well related to children maladjustment, independently of cultural differences.

Certainly, the relationship between parental control and adjustment is an important topic in parenting literature, although this manuscript has major gaps that it would have to improve to be able to contribute in this debate.

In the first references I have found that neither Baumrind [1] nor Maccoby and Martin [2] use monitoring label as part of parenting “control”. Maccoby and Martin said “Demanding, controlling – Undemanding, low in control attempts” p. 39, and Baumrind [1] don’t use the monitoring label in any part of text. Conceptually, Baumrind’ parenting control concept seems as need for the lack of children self-control [1].

Precisely, one of Maccoby and Martin's criticisms of previous studies [3] was that they analyzed disciplinary techniques one by one with children outcomes [3,4] without considering that “The interacting effects of control and warmth clearly differ from the interacting effects of restrictiveness and warmth” [1] p. 131. In same line, Baumrind conclusion is that “The spontaneity, warmth, and zest of Pattern I children were not affected adversely by high parental control.” [1] p. 132. In this point Maccoby and Martin's said “Readers will note that whereas one of the traditional dimensions of disciplinary techniques— power assertiveness—can be subsumed fairly well under the heading of authoritarian parenting, there is no place in the diagram for induction or other forms of reasoning, for causal attributions made by parents, or for withdrawal of love. We have clustered these aspects of parenting loosely in a separate section” [2] p. 39.

As authors recognize, parenting styles largely follow a four-typology model of parental socialization styles with two orthogonal dimensions (e.g., Darling & Steinberg, 1993, pp. 491-492 [4]; Smetana, 1995, p. 299 [5]; Steinberg, 2005, p. 71 [6]). These dimensions mirrored traditional parenting dimensions of warmth and strictness [7,8], as "responsiveness was often operationalized using measures of parental warmth and acceptance, while demandingness came to be defined with respect to parental firmness" (Steinberg, 2005, p. 71 [6]).

Although is well-know that, beyond the different labels, parenting psychological control is clearly related, as a dimension, to children maladjustment [9-11]. There are any empirical evidence that authors (of questionnaires? [12-13]) typically understand parental control closer to psychological control? (Lines 41-43). Have authors prove that all this control measures, as psychological control, are positive related to control or imposition and negatively related to warm or affection (i.e., rejection or hostility)?

I have found empirical evidence to the contrary (e.g., García & Gracia, 2009, p.114 [15]; Garcia, Serra, Zacares, & Garcia, 2018, p.157 [16]; Martinez, Garcia, Fuentes, Veiga, Garcia, Rodrigues, Cruise, & Serra, 2019, p.8  [17]), two parenting dimensions measures, warmth and strictness, seem modestly correlated. In line with orthogonality assumption [4-6] (R^2 correlations range from 0 to 1).

References

1. Baumrind, D. (1967). Child care practices anteceding three patterns of preschool behavior. Genetic Psychology Monographs, 75, 43-88.

2. Maccoby, E. E., & Martin, J. A. (1983). Socialization in the context of the family: Parent-child interaction. In P. H. Mussen (Ed.), Handbook of child psychology (Vol. 4, pp. 1-101). New York: Wiley.

3. Becker, W. C. Consequences of different kinds of parental discipline. In M. L. Hoffman & L. W. Hoffman (Eds.), Review of child development research (Vol. 1). New York: Russell Sage Foundation, 1964.

4. Darling, N., & Steinberg, L. (1993). Parenting style as context: An integrative model. Psychological Bulletin, 113, 487-496. doi:10.1037/0033-2909.113.3.487

5. Smetana, J. G. (1995). Parenting styles and conceptions of parental authority during adolescence. Child Development, 66, 299-316.

6. Steinberg, L. (2005). Psychological control: Style or substance? In J. G. Smetana (Ed.), New directions for child and adolescent development: Changes in parental authority during adolescence (pp. 71-78). San Francisco: Jossey- Bass.

7. Sears, R. R., Maccoby, E. E., & Levin, H. (1957). Patterns of child rearing. Evanston, IL: Row, Peterson.

8. Schaefer, E. S. (1959). A circumplex model for maternal behavior: Journal of Abnormal and Social Psychology, 59, 226-235.

9. Barber, B. K. (1996). Parental psychological control: Revisiting a neglected construct. Child Development, 67, 3296-3319. doi:10.1111/j.1467-8624.1996.tb01915.x

10. León-del-Barco, B., Mendo-Lázaro, S., Polo-del-Río, M. I., & López-Ramos, V. M. (2019). Parental psychological control and emotional and behavioral disorders among Spanish adolescents. International Journal of Environmental Research and Public Health, 16(507), 1-13. doi:10.3390/ijerph16030507

11. Tur-Porcar, A. M., Jiménez-Martínez, J., & Mestre-Escrivá, V. (2019). Substance use in early and middle adolescence. The role of academic efficacy and parenting. Psychosocial Intervention. doi:10.5093/pi2019a11

12. Rohner, R. P. (1989). Parental Acceptance-Rejection I Control Questionnaire (PARQI Control). Storrs, CT, 06268-1425: Rohner Research, 255 Codfish Falls Road.

13. Musitu, G., & Garcia, F. (2001). ESPA29: parental socialization scale in adolescence. Madrid, Spain: Tea.

14. García, F., & Gracia, E. (2009). Is always authoritative the optimum parenting style? Evidence from Spanish families. Adolescence, 44(173), 101-131.

15. Garcia, O. F., Serra, E., Zacares, J. J., & Garcia, F. (2018). Parenting styles and short- and long-term socialization outcomes: A study among Spanish adolescents and older adults. Psychosocial Intervention, 27, 153-161. doi:10.5093/pi2018a21

16. Martinez, I., Garcia, F., Fuentes, M. C., Veiga, F., Garcia, O. F., Rodrigues, Y., Cruise, E., & Serra, E. (2019). Researching parental socialization styles across three cultural contexts: Scale ESPA29 bi-dimensional validity in Spain, Portugal and Brazil. International Journal of Environmental Research and Public Health, 16(197), 1-14. doi:10.3390/ijerph16020197

Reviewer 3 Report

Dear authors,

your article is really clear and well written.

Please check some sentences for minor spelling errors.

Good luck!

Reviewer 4 Report

Dear Editor;

Thank you for inviting me to review the manuscript titled " Measurement and Function of the Control Dimension 2 in Parenting Styles: A Systematic Review", which is a well written review article that I think can contribute to the family research literature.  However, although the authors said whether parental control contributes positively or negatively to child development depends on which parenting scale they used to measure the concept. For this, I suggest the authors to more explain more clearly and comprehensively in terms of the contents and mechanism of different parenting scales in contribution to child development differently in the Discussion section.

Reviewer J

Jerf Yeung

Round 2

Reviewer 1 Report

Thank you to the authors for have provided a new version and a covering letter. I think the manuscript has improved.

However, I would like to make two basic comments:

1)    I think it's very important to write in the article about the orthogonality. This is a central notion. The orthogonzality has been argued and empirical proven by some authors such as Fernando García.

2)    It is very important to explain the search criteria in more detail, as in the cover letter.

I look forward to hearing your comments.